# Berries and Their Active Compounds in Prevention of Age-Related Macular Degeneration

**DOI:** 10.3390/antiox13121558

**Published:** 2024-12-18

**Authors:** Xiang Li, Lingda Zhao, Bowei Zhang, Shuo Wang

**Affiliations:** School of Medicine, Nankai University, Tianjin 300071, China; 2112340@mail.nankai.edu.cn (X.L.); 2120201373@mail.nankai.edu.cn (L.Z.)

**Keywords:** age-related macular degeneration (AMD), berries, phytochemicals, flavonoids

## Abstract

Age-related macular degeneration (AMD) is a leading cause of vision loss in the elderly, significantly diminishing quality of life. Currently, there is no available treatment to reverse retinal degeneration and neuronal loss, prompting a focus on interventions that slow the progression of intermediate AMD and geographic atrophy. Berries are rich in bioactive compounds, including flavonoids, anthocyanins, carotenoids, and resveratrol, known for their antioxidant, anti-inflammatory, and anti-angiogenic properties. Preclinical studies suggest that extracts from various berries, such as aronia, honeysuckle, black currant, goji, and bilberry, can improve retinal health by reducing oxidative stress and inflammation. Although clinical trials are limited, emerging evidence indicates that dietary intake of these compounds may enhance visual function and slow the progression of AMD. This review summarizes findings from both animal studies and clinical trials to identify specific berries that have been validated to prevent or delay AMD progression, as well as those with potential therapeutic value. Furthermore, we examine the key phytochemicals present in these berries, their mechanisms of action on macular degeneration, and their distinct properties for therapeutic application. A deeper understanding of these characteristics could enable the rational appliance of berries, especially wolfberry, and berry-derived components, such as carotenoids and anthocyanins, to optimize better therapeutic outcomes in AMD management.

## 1. Introduction

### 1.1. Definition and Classification of Age-Related Macular Degeneration

Age-related macular degeneration (AMD) accounts for 87% of all blindness globally and is the most common cause of blindness in developed countries, especially among people aged 60 and above [1]. AMD remains a major public health issue, and it is estimated that the number of individuals affected by AMD will soar to 288 million by 2040. AMD primarily erodes the macula, the central part of the retina responsible for sharp and detailed vision [2,3]. The pathogenesis of AMD is multifactorial, involving factors like oxidative stress, chronic inflammation, and dysregulated lipid metabolism. Currently, treatments aim to slow the progression of the disease rather than reverse retinal degeneration [3,4]. AMD is classified into two main forms: dry (atrophic) and wet (neovascular). Dry AMD progresses slowly and results in gradual central vision loss due to drusen accumulation and geographic atrophy. In contrast, wet AMD, characterized by rapid progression and severe vision loss, results from abnormal blood vessel growth and leakage. Patients typically seek medical attention when they notice significant vision problems [5]. The neurodegenerative process in both AMD and glaucoma causes irreversible retinal damage, making the focus on slowing disease progression crucial [3,4,5,6]. Early detection and intervention, especially in the transition from dry to wet AMD, are critical for preserving vision [7,8].

### 1.2. Berries and Phytochemicals

Berries, including blueberry, cranberry, currant, raspberry, and blackberry, are a wide group of blue, purple, or red small-sized and highly perishable fruits rich in phytochemicals such as anthocyanins, flavonoids, carotenoids, and resveratrol [9,10,11] (Figure 1). These compounds have gained significant attention for their health benefits, particularly in the prevention and management of chronic diseases, including diabetes, obesity, and hypertension [9,11,12,13]. Phytochemicals in berries have been studied to determine their effects on cardiovascular health and their potential to reduce heart disease risks [10,14,15]. Although numerous studies suggest that dietary polyphenols and other phytochemicals can slow the progression of AMD and other ocular diseases, specific berries and their respective components have not been systematically studied to determine their effects on macular degeneration [16,17,18]. Furthermore, there has been very little research on the combined use of different berries and effective components. This review aims to comprehensively analyze the current research on the effects of berries rich in phytochemicals on AMD and summarize several experiments on combined use and explore the future possibilities in this area.

## 2. Pathophysiology of AMD

### 2.1. Pathological Features

Oxidative stress is one of the major causes of AMD, leading to the formation of drusen—lipoprotein and inflammatory deposits within the retinal layers [19,20]. As the drusen enlarges, the condition progresses to early or intermediate AMD, depending on the size, number, and location of the drusen, along with retinal pigment epithelium (RPE) changes [21]. Geographic atrophy (GA), which is characterized by the degeneration of retinal pigment epithelial (RPE) cells, photoreceptors, and choroidal capillaries, may develop in late-stage AMD [19]. Drusen and pigmentary disturbances are key indicators of disease progression [4]. Advanced AMD may result in choroidal neovascularization (CNV), leading to the wet form of the disease. Abnormal blood vessel growth associated with CNV can cause scarring and leakage, leading to serious vision loss. Recent research has emphasized the critical role of the choroid in the pathobiology of AMD [3]. They proposed a revised nomenclature for neovascular complexes based on their anatomical localization via multimodal imaging, including optical coherence tomography (OCT). Like occult lesions, type 1 lesions show neovascularization beneath the RPE. Like classic lesions, type 2 neovascularization starts in the choroid and spreads throughout the sub-retinal region between the RPE and the neurosensory retina. Intraretinal neovascularization, which is comparable to retinal angiomatous proliferation, is a characteristic of type 3 lesions [3,22]. CNV can further lead to serious complications such as retinal detachment, vitreous hemorrhage, and fibrosis, which significantly impair vision [7]. Understanding these stages and their respective characteristics is crucial for diagnosing and managing AMD effectively.

### 2.2. Mechanisms of Development

Several pathways have been implicated in the gradual progression of vision loss in geographic atrophy (GA). Although the precise cause of AMD is still unknown, oxidative damage, persistent inflammation, and an excessive buildup of lipofuscin are probable contributing factors [23,24] (Figure 2).

The choriocapillaris, Bruch’s membrane, the retinal pigment epithelium (RPE), and photoreceptors all undergo morphological and physiological alterations as a result of accumulated oxidative damage brought on by aging. Autofluorescent lipoproteins called lipofuscin, which is mostly made up of indegradable remnants of the visual cycle, are among the intracellular detritus that RPE cells progressively collect. It is believed that this accumulation results in anomalies in extracellular matrix deposition, cell damage, and the disruption of RPE function. As a result, between the inner collagenous layer of Bruch’s membrane and the basal lamina of the RPE, lipid-rich membranous debris that contains RPE cell fragments, lipids, minerals, and immune system-associated materials (such as complement cascade components) builds up [3]. This material can accumulate locally in Drusen, the hallmark lesion of AMD, or it might deposit in a thin layer called basal linear deposits [25].

AMD is also associated with immune signaling, cell trafficking, and inflammation, with the complement cascade being the most extensively studied immune pathway in AMD [19]. Components derived from drusen are believed to activate the complement cascade, contributing to the progression of pathological changes in AMD [3]. It is believed that increased complement activation causes the choriocapillaris to degenerate by losing endothelial cells [26]. The choriocapillaris’s subsequent degradation increases inflammation, accelerates the course of AMD, and worsens oxidative damage to the outer retinal structures and the RPE that covers them. Ultimately, as the cumulative oxidative injury from hypoxia and chronic inflammation overwhelms the RPE’s ability to cope [27], this pathological cascade culminates in photoreceptor cell death.

Neovascularization is the process by which existing blood capillaries give rise to new capillary vessels. The primary cause of substantial vision loss in AMD patients is choroidal neovascularization (CNV), which can occur in a variety of chorioretinal illnesses, including AMD [3]. VEGF plays a pivotal role in the pathogenesis of CNV as a potent angiogenic factor. New blood vessels are created as a result of VEGF’s promotion of endothelial cell migration and proliferation. Its overexpression in the retinal pigment epithelium (RPE) is known as a key driver of CNV development. Under hypoxic conditions, HIF-1α is stabilized and activates the transcription of VEGF, which in turn binds to its receptors VEGFR1 and VEGFR2 on endothelial cells, initiating a cascade of signaling events that promote angiogenesis. Targeting VEGF has emerged as a key component in the treatment of CNV linked to AMD, as evidenced by the discovery that decreased expression levels of MMP-9 and VEGF relieve CNV generation in animal models by lowering the HIF-1α/VEGF/VEGFR2 pathway [28].

## 3. Intervention of Polyphenols Rich in Berries on AMD

Despite the current lack of effective measures to cure early age-related macular degeneration (AMD), other than blue-light protection, there is a growing interest in dietary interventions. For example, the Mediterranean-type diet was reported to be associated with a decreased risk of progression to late AMD, the major components of which include minerals, vitamins, omega-3 fatty, and carotenoids [29]. Higher dietary intake of vitamin supplementation has also been known to significantly reduce the risk of progression of AMD in the Age-Related Eye Disease Study (AREDS) 1 and 2 and has been widely accepted as a preventive treatment for AMD [18,30]. In particular, flavonoids, anthocyanins, resveratrol, and carotenoids, which are naturally occurring compounds rich in berries, have attracted attention for their potential role in preventing the progression of AMD [31,32,33]. In the following sections, we will delve deeper into the specific roles of these phytochemicals and their mechanisms of action.

It is important to note that our literature review included a comprehensive search of common phytochemicals in PubMed. We focused on key categories such as flavonoids, phenolic acids, lignans, stilbenes, carotenoids, volatile terpenoids, alkaloids, and isothiocyanates. In this extensive search, we selected only those phytochemicals that had sufficient supporting research to warrant a detailed discussion in the context of AMD prevention and management.

### 3.1. Flavonoids

Flavonoids are a prominent subclass of polyphenolic compounds that encompass a wide variety of structurally related molecules. This diverse group includes flavonols, flavones, flavanols, flavanones, and isoflavonoids. While anthocyanins are technically part of the flavonoid family, they are often discussed separately due to their unique properties and significant therapeutic potential. Structurally, all flavonoids share a common chemical backbone consisting of two aromatic rings connected by a three-carbon bridge (C6-C3-C6), typically forming an oxygenated heterocyclic C-ring. Variations in the structure of this heterocyclic C-ring result in the formation of different flavonoid subclasses, such as flavonols, flavones, flavanols, flavanones, anthocyanidins, and isoflavonoids [34]. Flavonoids are widely distributed in the plant kingdom and are abundant in many dietary sources. For example, tea, onions, and apples are particularly rich in flavonoids, though they can also be found in a broad range of colorful fruits and vegetables, each offering varying concentrations and types of these bioactive compounds [35].

The potential role of flavonoids in the prevention and management of AMD has attracted significant scientific interest due to their potent antioxidant, anti-inflammatory, and anti-angiogenic properties (Figure 3) [11]. In terms of their antioxidant effects, flavonoids are known to activate the nuclear factor erythroid 2-related factor 2 (Nrf2) signaling pathway, which leads to the upregulation of key antioxidant enzymes such as heme oxygenase-1 (HO-1), NAD(P)H quinone dehydrogenase 1 (NQO-1), glutamate-cysteine ligase modifier subunit (GCLM), glutamate-cysteine ligase catalytic subunit (GCLC), superoxide dismutase (SOD), and glutathione (GSH). This activation helps mitigate oxidative stress, which is a critical factor in the pathogenesis of AMD [20]. Additionally, flavonoids have been shown to modulate the phosphoinositide 3-kinase/protein kinase B (PI3K/AKT) signaling pathway, regulating the expression of apoptosis-related proteins such as Bax, Bcl-2, and caspase-3. This modulation helps inhibit the apoptosis of RPE cells, thereby preventing or slowing down the progression of AMD [20].

Notably, dietary flavonoids such as fisetin, luteolin, quercetin, eriodictyol, baicalein, galangin, and epigallocatechin gallate (EGCG), along with synthetic flavonoid analogs like 3,6-dihydroxy flavonol and 3,7-dihydroxy flavonol, have shown protective effects against oxidative stress-induced cell death in human RPE cells [27]. These compounds enhance cellular antioxidant defenses by upregulating phase 2 detoxification enzymes under the regulation of the Nrf2 pathway, thereby providing a robust defense against oxidative damage.

In addition to their antioxidant properties, flavonoids also exhibit strong anti-inflammatory effects. They achieve this by inhibiting key signaling pathways, including Toll-like receptor 4/nuclear factor kappa-light-chain-enhancer of activated B cells (TLR4/NF-κB), mitogen-activated protein kinase (MAPK), and nucleotide-binding oligomerization domain-like receptor family pyrin domain-containing 3 (NLRP3) inflammasome pathways. This inhibition leads to a reduction in the expression of pro-inflammatory mediators such as interleukin-8 (IL-8), intercellular adhesion molecule 1 (ICAM-1), matrix metalloproteinase-9 (MMP-9), monocyte chemoattractant protein-1 (MCP-1), interleukin-1 beta (IL-1β), and interleukin-6 (IL-6). The suppression of these inflammatory pathways contributes to the therapeutic potential of flavonoids in managing AMD and other inflammation-related ocular conditions.

Furthermore, flavonoids have demonstrated anti-angiogenic properties, which are particularly relevant in the treatment of wet AMD, a form of disease characterized by abnormal blood vessel growth under the retina. By downregulating the hypoxia-inducible factor 1-alpha/vascular endothelial growth factor/vascular endothelial growth factor receptor 2 (HIF-1α/VEGF/VEGFR2) signaling pathway, flavonoids effectively reduce the expression of critical mediators of choroidal neovascularization (CNV) [28,36]. This action helps to prevent the formation of new, fragile blood vessels that can leak and cause vision loss in patients with wet AMD.

This multifaceted action of flavonoids—encompassing antioxidative, anti-inflammatory, and anti-angiogenic effects—underscores their potential as therapeutic agents in both the prevention and management of AMD. However, it is important to note that the bioavailability of these compounds, as well as their clinical efficacy, remains an area that requires further investigation. Understanding how flavonoids are absorbed, metabolized, and utilized in the body is crucial for developing effective interventions that can maximize their therapeutic potential in AMD treatment.

### 3.2. Anthocyanins

Anthocyanins, a specific subclass of flavonoids, are the pigments responsible for the vivid red, blue, and purple colors observed in many berries, such as blueberries, blackcurrants, and bilberries. These compounds are widely recognized for their potent antioxidative properties, which play a crucial role in the maintenance of overall health, including ocular health [37]. The antioxidative capacity of anthocyanins is particularly important for protecting retinal cells from oxidative stress, a key factor implicated in the development and progression of AMD [38]. By scavenging ROS, anthocyanins help to neutralize harmful oxidative molecules, thereby mitigating cellular damage and preserving the integrity and function of retinal cells.

In addition to their strong antioxidative properties, anthocyanins also exhibit significant anti-inflammatory effects, which further contribute to their potential role in AMD prevention and management. These compounds modulate inflammatory responses by inhibiting key signaling pathways, such as the mitogen-activated protein kinase (MAPK) pathways, including extracellular signal-regulated kinase (ERK1/2) and p38 MAPK, which are commonly associated with the activation of inflammatory processes [13,39]. Furthermore, anthocyanins activate the phosphoinositide 3-kinase/protein kinase B (PI3K/Akt) pathway, a crucial cellular signaling pathway that enhances the antioxidant defense mechanisms of cells, thereby providing additional protection against oxidative stress and inflammation [40]. These anti-inflammatory actions reduce the expression of pro-inflammatory cytokines, such as interleukins and tumor necrosis factor-alpha (TNF-α), which are known to contribute to inflammation-induced retinal damage. Consequently, anthocyanins help protect retinal cells from the harmful effects of chronic inflammation, making them particularly beneficial in the prevention and management of AMD [41].

Moreover, anthocyanins have been shown to directly influence retinal function and support visual performance. Research on berries such as bilberries and blackcurrants has demonstrated that anthocyanins can enhance visual function by preserving retinal integrity, reducing oxidative stress, and improving microcirculation within the retina [38,42]. Improved retinal microcirculation is essential for delivering oxygen and nutrients to retinal cells, thereby supporting their function and survival. These effects contribute to the maintenance of healthy vision and may slow the progression of AMD, particularly in its early stages, when intervention is most effective.

Despite the promising effects of anthocyanins, their clinical application faces significant challenges, primarily related to their bioavailability. Bioavailability refers to the extent and rate at which a compound is absorbed into the bloodstream and made available to target tissues, such as the retina. Anthocyanins are rapidly metabolized and excreted from the body, which limits their concentration and effectiveness in retinal tissues. This rapid clearance poses a barrier to achieving the desired therapeutic outcomes in ocular health [41].

To overcome these challenges, further research is needed to develop strategies that enhance the bioavailability of anthocyanins. One potential approach is the use of advanced delivery systems, such as encapsulation in liposomes or nanoparticles, which can protect anthocyanins from degradation in the gastrointestinal tract and facilitate their absorption into the bloodstream. Additionally, combining anthocyanins with other bioactive compounds or dietary components that enhance their stability and absorption may also improve their bioavailability. For example, piperine, a compound found in black pepper, has been shown to enhance the bioavailability of various phytochemicals by inhibiting their metabolism and increasing their absorption [3]. Exploring such combination therapies could be a promising avenue for maximizing the therapeutic potential of anthocyanins in AMD prevention and treatment.

Moreover, more robust clinical trials are required to establish standardized dosages and treatment protocols for anthocyanin supplementation. Current clinical studies vary widely in terms of the doses used, duration of treatment, and specific anthocyanin sources, making it difficult to draw definitive conclusions about their efficacy in AMD management. Well-designed, large-scale clinical trials that account for these variables are necessary to determine the optimal dosage and treatment regimens needed to achieve significant clinical benefits. Establishing such standards will be crucial for translating promising preclinical and observational findings into effective clinical practice.

In summary, anthocyanins possess a range of beneficial properties, including antioxidative, anti-inflammatory, and vision-enhancing effects, making them a promising candidate for the prevention and management of AMD. However, their clinical application is currently limited by challenges related to bioavailability and the need for standardized treatment protocols. Future research aimed at overcoming these barriers will be essential to fully harness the potential of anthocyanins as a therapeutic option for AMD and other retinal disorders.

### 3.3. Carotenoids

Carotenoids are a diverse group of naturally occurring pigments found in a wide variety of fruits and vegetables, as well as in plants, algae, and photosynthetic bacteria. These pigments are responsible for the vibrant red, orange, and yellow hues seen in many plant-based foods. Since humans are unable to synthesize carotenoids on their own, they must obtain these essential nutrients through diet or supplementation. Carotenoids are known to play multiple roles in human health, including acting as antioxidants, inhibiting the growth of malignant tumors, and inducing apoptosis in cancer cells [43]. The primary mechanism by which carotenoids exert their health benefits is through their potent antioxidant activity, which helps neutralize harmful free radicals in the body. However, they also influence various biological pathways that contribute to their protective effects. For instance, β-carotene serves as a pro-vitamin A, which is essential for vision, immune function, and skin health, while lutein and zeaxanthin are key components of the macular pigment in the eye, providing targeted protection for retinal tissues [44].

Carotenoids, particularly lutein and zeaxanthin, play a crucial role in maintaining eye health. Their primary function in the eye is to absorb blue light and protect the retina from photochemical injury, which is a major risk factor for AMD [45]. Lutein and zeaxanthin are concentrated in the macula, the central part of the retina responsible for high-resolution vision. As local antioxidants, they reduce the formation of lipofuscin, a harmful byproduct of oxidative stress, and minimize oxidative damage to retinal cells [46]. Furthermore, these carotenoids have been shown to inhibit the activation of the nuclear factor kappa-light-chain-enhancer of activated B cells (NF-κB) and mitogen-activated protein kinase (MAPK) pathways, both of which are involved in the inflammatory response. By modulating these pathways, lutein and zeaxanthin help reduce inflammation in the retina, which is a contributing factor in the progression of AMD.

Another critical pathway influenced by carotenoids is the nuclear factor erythroid 2-related factor 2 (Nrf2) pathway. This pathway plays a vital role in regulating the expression of antioxidant enzymes that enhance the cellular defense system against oxidative stress. By activating the Nrf2 pathway, carotenoids promote the regeneration and repair of retinal cells, further supporting eye health and reducing the risk of AMD progression. Additionally, β-carotene, one of the most studied carotenoids, can be converted into vitamin A in the body. Vitamin A is essential for maintaining good vision, particularly in low-light conditions, and supports the overall health of the retina by promoting the function and renewal of photoreceptor cells.

The effects of lutein and zeaxanthin supplementation on both early and late stages of AMD have been extensively investigated. While many studies support their beneficial role in slowing AMD progression, inconsistencies remain in the findings, particularly concerning the optimal dosage and long-term outcomes of supplementation. These discrepancies may be attributed to variations in study design, differences in population characteristics, and the use of diverse supplementation regimens. A more critical analysis of these studies is necessary to resolve these inconsistencies and provide clearer guidance on the effective use of lutein and zeaxanthin for AMD prevention and management. Such an analysis would help refine clinical guidelines and dietary recommendations, ensuring that individuals at risk of or suffering from AMD receive the most effective nutritional support.

A notable study, the Age-Related Eye Disease Study 2 (AREDS2), demonstrated the beneficial effect of adding lutein and zeaxanthin to the original Age-Related Eye Disease Study (AREDS1) formulation at a ratio of 5:1 [6,18]. The inclusion of these carotenoids was shown to enhance the formulation’s effectiveness in supporting macular health and reducing the risk of AMD progression. Moreover, the AREDS2 provided further insights into the role of carotenoids in AMD management. This large-scale clinical trial found that eliminating beta-carotene or using lower doses of zinc did not significantly affect the progression to advanced AMD. However, supplementation with lutein and zeaxanthin showed a potential beneficial association with slowing the progression of late-stage AMD [18]. These findings underscore the importance of lutein and zeaxanthin in maintaining retinal health and suggest that they may be more effective and safer alternatives to beta-carotene in AMD prevention, particularly in populations at risk for lung cancer, where beta-carotene supplementation has been associated with increased risk.

Despite the strong evidence supporting the benefits of carotenoids in AMD, several challenges remain in optimizing their delivery and absorption in the human body. The bioavailability of these compounds, which refers to the extent and rate at which they are absorbed and utilized in the body, can vary significantly depending on the source and formulation. For example, carotenoids from food sources are often more bioavailable than those from supplements due to the presence of other dietary components that aid in their absorption. Factors such as the food matrix, the presence of dietary fats, and individual differences in metabolism can all influence how well carotenoids are absorbed and reach target tissues like the retina.

To maximize the protective effects of carotenoids in AMD, further research is needed to develop more effective supplementation strategies. This includes exploring new delivery systems, such as nanoencapsulation, which can enhance the stability and absorption of carotenoids. Additionally, combining carotenoids with other nutrients or bioactive compounds that synergistically enhance their effects could be a promising approach. For example, combining lutein and zeaxanthin with omega-3 fatty acids has been shown to improve their bioavailability and efficacy in supporting eye health. Future studies should focus on optimizing these formulations and determining the most effective dosages and combinations to ensure that individuals receive the maximum benefit from carotenoid supplementation in the context of AMD prevention and treatment.

### 3.4. Resveratrol

Resveratrol (3,5,4′-trihydroxy-trans-stilbene) is a type of stilbenoid, a natural phenolic compound, and a phytoalexin produced by various plants in response to injury, ultraviolet radiation, or pathogen attack. It is primarily found in the skins of grapes, blueberries, raspberries, and mulberries and is the principal biologically active component in red wine. The compound gained significant attention due to the “French Paradox”, which observed lower rates of cardiovascular disease in the French population despite a diet high in saturated fats. This paradox was attributed, at least in part, to the regular consumption of red wine, suggesting that its resveratrol content might confer cardiovascular benefits [47]. Since then, interest in resveratrol has surged, leading to extensive research into its potential health benefits.

Resveratrol is well-recognized for its potent antioxidant and anti-inflammatory properties and has been shown to offer cardioprotective, neuroprotective, and anti-aging benefits [48,49]. These effects have been widely studied in the context of various chronic conditions, but the extent to which resveratrol’s benefits translate to clinical outcomes in AMD remains relatively unexplored. To better understand resveratrol’s potential role in AMD management, further studies are required to elucidate its pharmacokinetics specifically in the context of AMD. This includes understanding how resveratrol is absorbed, distributed, metabolized, and excreted in the body, as well as determining its long-term safety and efficacy when used as part of a dietary intervention.

As an antioxidant, resveratrol has been shown to enhance mitochondrial bioenergetics, thereby improving the efficiency of cellular energy production and reducing the generation of ROS [50]. In human retinal RPE cells, resveratrol effectively reduces ROS production, helping to protect these cells from oxidative damage that is a key factor in the pathogenesis of AMD [2,51]. It also offers protection against UVA-induced damage in ARPE-19 cells, a human cell line frequently used to study retinal diseases [52]. These protective effects are partly mediated through the activation of sirtuin 1 (SIRT1), a histone deacetylase enzyme that plays a crucial role in regulating cellular stress responses, longevity, and metabolic function [53]. Resveratrol can directly and indirectly activate SIRT1, leading to the modulation of various downstream pathways involved in cellular protection and repair.

Additionally, resveratrol inhibits cyclic adenosine monophosphate (cAMP)-degrading phosphodiesterase, which results in elevated cAMP levels, activation of the exchange protein directly activated by cAMP (Epac1), and subsequent stimulation of the calcium/calmodulin-dependent protein kinase beta (Camkβ) and AMP-activated protein kinase (AMPK) pathways. This sequence of events increases the levels of nicotinamide adenine dinucleotide (NAD+), further enhancing SIRT1 activity and promoting cellular homeostasis [54]. Through these mechanisms, resveratrol supports the survival of retinal cells and helps to maintain their function under conditions of oxidative stress and inflammation.

Resveratrol has also been shown to modulate other molecular targets such as DNA methyltransferases (DNMTs), which play a role in gene expression and epigenetic regulation. By restoring the methylation levels of long interspersed nuclear element-1 (LINE-1), a marker of aging, resveratrol can induce autophagy, promote cell survival, and reduce inflammation in human cells [55,56,57,58]. It can also inhibit macrophage infiltration and downregulate the expression of inflammatory and angiogenesis-related factors, such as vascular endothelial growth factor (VEGF), intercellular adhesion molecule 1 (ICAM-1), and monocyte chemoattractant protein-1 (MCP-1). These actions are particularly relevant in the context of AMD, where chronic inflammation and pathological neovascularization are central to disease progression.

Furthermore, resveratrol has demonstrated protective effects against A2E-induced damage in ARPE-19 cells. A2E is a major component of lipofuscin, a toxic byproduct of the visual cycle that accumulates in RPE cells and contributes to retinal degeneration [59,60]. By preventing the accumulation of lipofuscin and reducing VEGF-A secretion, resveratrol shows potential in preventing the development of wet AMD, which is characterized by abnormal blood vessel growth in the retina [61]. Studies in animal models have further supported these findings, showing that resveratrol alleviates choroidal neovascularization (CNV) by modulating the hypoxia-inducible factor 1-alpha (HIF-1α)/VEGF/VEGFR2 pathway [28]. Resveratrol exerts its effects on various cell types in the eye, including RPE cells, vascular endothelial cells, and macrophages, by inhibiting NF-κB activation, reducing inflammation, and preventing angiogenesis [62].

Moreover, resveratrol has been reported to inhibit pathological retinal neovascularization in animal models, such as mice [63]. These findings suggest that resveratrol may offer therapeutic potential not only in AMD but also in other retinal diseases characterized by abnormal blood vessel growth and inflammation.

Although clinical trials specifically investigating the effects of resveratrol on AMD are limited, the available studies show promising results. For instance, an over-the-counter (OTC) oral supplement containing microencapsulated resveratrol, red wine polyphenols, and other compounds (L/RV) was found to improve the health of the retinal-RPE-choroidal complex in a clinical trial [64]. Another study reported improvements in RPE function and choroidal blood flow in AMD patients following resveratrol-based nutritional supplementation, suggesting its potential efficacy in AMD management [65]. Long-term follow-up studies have shown broad bilateral improvements in ocular structure and function in AMD patients, further supporting the use of resveratrol as a component of AMD treatment [64].

Despite these promising findings, the clinical application of resveratrol in AMD treatment faces significant challenges. One of the primary obstacles is its low bioavailability, which means that high doses are required to achieve therapeutic effects [65]. Resveratrol is rapidly metabolized and excreted, limiting its ability to reach and maintain effective concentrations in retinal tissues. To address this issue, current research is focused on developing more bioavailable formulations, such as nanoparticle-based delivery systems, and exploring synergistic combinations with other compounds to enhance their efficacy. For instance, combining resveratrol with other antioxidants or anti-inflammatory agents may help to amplify its protective effects.

In conclusion, while resveratrol shows considerable promise as a therapeutic agent for AMD, further long-term clinical trials are necessary to establish standardized treatment protocols and to fully understand its potential in managing this complex condition. Future research should focus on optimizing resveratrol delivery and absorption, as well as identifying the most effective dosages and combinations for use in AMD prevention and treatment.

## 4. Experimental and Clinical Studies of Different Berries

Over recent years, a growing body of epidemiological and clinical studies has highlighted the protective effects of berries against a wide range of non-communicable chronic diseases, such as cardiovascular diseases, diabetes, and neurodegenerative disorders [66]. The consumption of berries has been associated with a reduction in all-cause mortality, as documented in a comprehensive meta-analysis by Aune et al. [67]. Additionally, the risk of developing age-related macular degeneration (AMD) can potentially be mitigated through the consumption of antioxidant-rich foods, dietary supplements, and nutraceutical formulas designed to support eye health [68]. The available data in the literature also support the potential role of berries in AMD, which contain elements that are key players in these supplements and dietary patterns [29,30]. This section aims to provide a comprehensive overview of studies investigating the impact of different types of berries on AMD, categorizing them into two groups: alternative berries, which have shown promising effects in treating or preventing AMD, and potential berries, which are currently being explored for their therapeutic properties.

### 4.1. Available Berry Options

Certain berries, such as bilberry, have long been used in traditional medicine to treat various eye conditions. Clinical trials have provided some evidence supporting their efficacy, but further research is necessary to confirm these findings, especially concerning long-term outcomes such as the effect on visual function and quality of life in individuals with AMD.

#### 4.1.1. Bilberry, *Vaccinium myrtillus* L.

Bilberry, which is particularly rich in anthocyanins, has been traditionally utilized due to its beneficial effects on eye health. Preclinical studies have indicated that bilberry extract may protect against retinal inflammation and oxidative stress. This is particularly significant because oxidative stress and inflammation are key factors in the development and progression of AMD. Bilberry has been shown to enhance the clearance of beta-amyloid deposits in drusen, a characteristic feature of AMD, and to inhibit the activation of STAT3 and NF-κB, which are pathways associated with inflammation and cell survival [8,42,69].

In the study conducted by Miyake et al., bilberry extract demonstrated a protective effect on visual function during retinal inflammation in mice, preventing the impairment of photoreceptor cells [42]. Furthermore, bilberry exposure was associated with the promotion of misfolded protein clearance, such as beta-amyloid, which accumulates in drusen, a hallmark of AMD pathology [69]. Additionally, bilberry extract has been found to mitigate endotoxin-induced uveitis by reducing rhodopsin depletion and preventing the shortening of the outer segments of photoreceptor cells. It also inhibits STAT3 activation, a pathway linked to inflammation-induced rhodopsin reduction, and decreases the expression of interleukin-6, a cytokine that activates STAT3 [42]. Beyond its anti-inflammatory properties, the anthocyanin-rich bilberry extract has shown the ability to reduce intracellular reactive oxygen species (ROS) and modulate the activation of NF-κB, a transcription factor sensitive to changes in cellular redox states, in the inflamed retina [8,42,70].

Clinical trials have also suggested that bilberry extract may help alleviate eye fatigue. For instance, studies have shown that it can improve the function of the ciliary muscle, which is responsible for focusing and upregulating antioxidant defense enzymes in the retinal pigment epithelium [71,72]. In a prospective, randomized, double-blind, placebo-controlled study, Ozawa et al. reported that bilberry extract (BE) improved both objective and subjective measures of eye fatigue induced by the prolonged use of video display terminals [72]. Similarly, Kosehira et al. found that a daily intake of 240 mg of standardized bilberry extract (SBE) over 12 weeks significantly alleviated ciliary muscle accommodation issues associated with extended near-vision tasks and visual display terminal usage [71]. Furthermore, Milbury et al. observed that anthocyanins and other phenolic compounds from bilberry could upregulate enzymes that defend against oxidative stress, such as heme-oxygenase-1 and GST-pi, in retinal pigment epithelium (RPE) cells, indicating a potential role in activating genes controlled by the antioxidant response element [73].

Despite these promising findings, most studies on bilberry focus primarily on its effects on visual fatigue rather than directly addressing AMD progression. Consequently, there is a pressing need for more robust, long-term clinical trials specifically targeting AMD patients to validate these preliminary findings.

#### 4.1.2. Aronia, *Aronia melanocarpa*

Aronia, also known as chokeberry, is a berry rich in anthocyanidins, which are compounds well-known for their potent antioxidative properties. Numerous preclinical studies have demonstrated that aronia extract can protect the retina from oxidative damage and improve the expression of retinal proteins, which may help prevent retinal degeneration and associated visual impairment [17,74]. Specifically, Xing et al. found that aronia fruit extract could protect the retina from damage induced by sodium iodate (NaIO_3_), a chemical commonly used to model retinal degeneration in experimental settings. The study confirmed that aronia has a significant antioxidative effect on both the retina and serum, suggesting that it can mitigate oxidative stress, a major contributing factor to retinal cell damage [74]. Furthermore, the researchers observed that aronia extract positively influenced the expression of crystallin proteins in the retina, which play a critical role in maintaining the stability and function of retinal nerve cells. This effect likely contributes to protecting retinal nerve cells from secondary degeneration, which is often a consequence of initial retinal damage.

Moreover, subsequent studies have investigated the synergistic effects of combining aronia extract with other bioactive substances. For instance, combining Lactobacillus fermentum NS9, a probiotic, with anthocyanidin extract from aronia was found to significantly alleviate NaIO_3_-induced retinal damage. This combination not only improved retinal crystallin expression and enhanced antioxidant capacity but also helped to restore healthy microbiota balance, a factor increasingly recognized as important in ocular health. Notably, the combined treatment was markedly more effective than using aronia anthocyanidin extract alone, indicating potential benefits from multi-modal dietary interventions [17].

In addition to its antioxidative effects, aronia also possesses noteworthy anti-inflammatory properties. Ohgami et al. reported that aronia crude extract exhibited a dose-dependent anti-inflammatory effect in ocular inflammation models. The researchers found that the extract directly inhibited the expression of inducible nitric oxide synthase (iNOS) and cyclooxygenase-2 (COX-2) enzymes, leading to the suppression of key inflammatory mediators such as nitric oxide (NO), prostaglandin E2 (PGE2), and tumor necrosis factor-alpha (TNF-α). In an animal model of endotoxin-induced uveitis (EIU), which is an acute inflammatory condition affecting the anterior segment of the eye, aronia crude extract was shown to significantly reduce inflammation in a dose-dependent manner [75]. These findings highlight aronia’s potential as an anti-inflammatory agent that could be beneficial in managing inflammatory eye conditions.

Despite these promising preclinical results, there are currently no clinical trials directly linking aronia to the prevention or treatment of AMD. However, some studies suggest potential benefits of aronia in other ocular conditions. For example, a recent study by Szumny et al. indicated that aronia might improve near visual acuity in individuals with presbyopia, a common age-related condition characterized by difficulty focusing on close objects [76]. To fully understand aronia’s potential in AMD management, it is essential to conduct human clinical trials that evaluate its impact on AMD progression, determine appropriate dosages, and assess long-term safety.

#### 4.1.3. Wolfberry, *Lycium barbarum* L.

Wolfberry, commonly known as goji berry, is renowned for its high content of bioactive compounds, including lutein, zeaxanthin, polysaccharides, betaine, and taurine, all of which are believed to contribute to its potential eye health benefits. Lutein and zeaxanthin, in particular, are carotenoids that play a crucial role in protecting the eyes from oxidative stress and high-energy light exposure. These compounds are primarily found in the macula, the part of the retina responsible for central vision, where they act as antioxidants and light filters. Wolfberries contain diester forms of these carotenoids, which may enhance their bioavailability and efficacy in supporting eye health.

The well-rounded mechanisms by which wolfberries may exert protective effects against eye diseases are multifaceted and have been extensively studied, including mitochondrial function, inflammation, apoptosis, and antioxidants, as well as inhibiting neuronal degeneration, so we summarized them in Table 1. For example, Bertoldi et al. demonstrated that consuming an average of 15 g of goji berries per day in warm water could provide an adequate daily intake of zeaxanthin, estimated at approximately 3 mg, which is beneficial for maintaining healthy eyes [77]. This dietary supplement could potentially increase the concentration of zeaxanthin in the retina, thereby enhancing the eye’s defense against oxidative damage.

In a randomized pilot trial, regular intake of goji berries was shown to significantly increase macular pigment optical density (MPOD), an indicator of the amount of lutein and zeaxanthin in the macula. Higher MPOD is associated with a lower risk of developing AMD, as these pigments help filter harmful blue light and combat oxidative stress in the retina [78]. Another study highlighted the neuroprotective effects of goji berries in retinitis pigmentosa, a genetic disorder that leads to progressive retinal degeneration. The trial suggested that goji berries could help slow down retinal degeneration and preserve visual function in affected individuals [79]. Despite these promising findings, the current body of clinical research on wolfberries and AMD is rather limited.

**Table 1 antioxidants-13-01558-t001:** Summary of animal experiments and functions of wolfberry.

Functions	Subjects	Diseases	Targets	Dose	Ref
Mitochondrial Function	mice	diabetic	upregulation of peroxisome proliferator-activated receptor γ co-activator 1α, nuclear respiratory factor 1, mitochondrial transcription factor A	1% (kCal) wolfberry, 56 days	[80]
Inflammation	mice	retinitis pigmentosa	downregulation of NF-κB and HIF-1α	1 mg/kg of polysaccharides, 25, 29, or 41 days	[81]
	retinal pigment epithelium cells	diabetic	upregulation of peroxisome proliferator-activated receptor-γ (PPAR-γ)	0.1, 0.5, and 1 mg/mL of wolfberry extract	[82]
	mice	retinitis pigmentosa	downregulation of NF-κB and HIF-1α	1 mg/kg of polysaccharides, 25, 29, or 41 days	[81]
Apoptosis	rats	retinitis pigmentosa	downregulation of N-methyl-N-nitrosourea (MNU), poly (ADP-ribose) polymerase (PARP), and caspase	100, 200, and 400 mg/kg of polysaccharides, 8 or 14 days	[83]
	retinal pigment epithelium cells	age-related macular degeneration	downregulation of Aβ1-40, NOD-like receptors protein 3 (NLRP3), caspase-1, membrane N-terminal cleavage product of GSDMD, endoplasmic reticulum stress	0.1 mg/mL, 0.25 mg/mL, 0.5 mg/mL, or 1 mg/mL of wolfberry extract	[84]
				3.5 mg/L and 14 mg/L of polysaccharides	[85]
	mice	ischemia–reperfusion (I/R)	upregulation of superoxide dismutase (SOD), glutathione peroxidase (GSH-Px), malondialdehyde, nuclear factor erythroid 2-related factor 2 (Nrf2), HO-1, thioredoxin reductase (TrxR1)	3.6 and 7.2 g/kg of wolfberry extract, 28 days	[86]
				150 mg/kg and 300 mg/kg of polysaccharides, 7 days	[87]
				1 mg/kg of polysaccharides, 7 days	[88]
	rats	glaucoma	downregulation of JNK/c-jun pathway	0.1 mg/kg, 1 mg/kg, or 10 mg/kg of polysaccharides, 14 or 21 days	[89]
Antioxidant	mice, rats	acute ocular hypertension	downregulation of RAGE, ET-1, Aβ, and AGE	1 mg/kg and 10 mg/kg of polysaccharides, 28 days	[90]
				1 mg/kg of polysaccharides, 11 days	[91]
	rats	glaucoma	inhibition of progressive loss of retinal ganglion cells	1 mg/kg of wolfberry extract	[92]
Inhibiting Neuronal Degeneration	rats	chronic ocular hypertension	downregulation of endothelin-1	1 mg/kg of polysaccharides, 14 days	[93]

This table provides a comprehensive overview of recent animal experiments and functions of wolfberry, investigating various biological functions and pathological conditions across different models and systems. The focus is on mitochondrial function, inflammation, apoptosis, antioxidant effects, and neuronal degeneration inhibition. Key molecular factors and pathways are highlighted for each condition, elucidating the underlying mechanisms and potential therapeutic targets.

#### 4.1.4. Black Currant, *Ribes nigrum* L.

Black currant (BC, *Ribes nigrum* L.) is a fruit known for its high content of anthocyanins, which are powerful bioactive compounds. The primary anthocyanins present in black currant include delphinidin-3-rutinoside (D3R), delphinidin-3-glucoside (D3G), cyanidin-3-rutinoside (C3R), and cyanidin-3-glucoside (C3G). These compounds are not only responsible for the deep purple color of the berries but also contribute significantly to their health benefits. Black currant and its anthocyanins exhibit various biological activities, such as anticancer properties, vascular protection, and anti-obesity effects [94]. This versatility makes black currant a promising candidate for various therapeutic applications, including the potential prevention and management of AMD.

Recent studies have demonstrated that black currant extracts can promote the regeneration of rhodopsin, a visual pigment crucial for low-light vision, and inhibit the accumulation of harmful compounds like N-retinyl-N-retinylidene ethanolamine (A2E), which is associated with retinal damage and AMD progression [95]. A study by Shin et al. revealed that black currant extract effectively inhibited the accumulation of A2E in ARPE-19 cells, a human RPE cell line commonly used in AMD research. The extract also significantly downregulated several genes that were upregulated by A2E and exposure to blue light, thus protecting the cells from the deleterious effects of blue light by rescuing the expression levels of superoxide dismutase 1 (SOD1), an enzyme involved in protecting cells from oxidative stress [95]. These findings suggest that black currant may help safeguard retinal health by preventing oxidative damage and the accumulation of toxic metabolites in the retina.

Moreover, earlier research by Matsumoto et al. highlighted that delphinidin-3-rutinoside, a specific anthocyanin found in black currant, causes relaxation of the bovine ciliary smooth muscle and exerts an inhibitory effect on endothelin-1-induced contraction. Endothelin-1 is a potent vasoconstrictor, and its overactivity can lead to increased intraocular pressure and potential damage to ocular tissues. The ability of delphinidin-3-rutinoside to counteract these effects suggests that black currant may play a role in maintaining ocular blood flow and reducing the risk of conditions like glaucoma, which, like AMD, can lead to vision loss [96].

In another study, Matsumoto and colleagues reported that anthocyanins from black currant fruits could induce the regeneration of rhodopsin in rod photoreceptors. Rhodopsin regeneration is critical for the maintenance of healthy night vision and overall visual function, particularly in low-light conditions. The study suggested that black currant anthocyanins might help preserve visual acuity and slow down the degeneration of photoreceptor cells, which are vital for converting light into neural signals [97].

In terms of clinical research, Kan et al. conducted a randomized, double-blind, placebo-controlled clinical trial to investigate the protective effects of a novel botanical combination that included lutein ester; zeaxanthin; and extracts of black currant, chrysanthemum, and goji berry on adults experiencing eye fatigue (Table 2). The study found that this botanical formula significantly increased macular pigment optical density (MPOD) at both 45 and 90 days compared to the placebo group, suggesting an enhancement in macular health and potential protection against conditions such as AMD [98]. Increasing MPOD is important because it is associated with a lower risk of developing AMD, as the macular pigments lutein and zeaxanthin help filter blue light and reduce oxidative damage in the retina. Despite these encouraging findings, further research is needed to specifically evaluate black currant’s effects on AMD progression. Most studies have focused on general eye health benefits or related conditions, and there is a lack of direct evidence connecting black currant consumption with reduced AMD risk or slowed disease progression.

### 4.2. Berries with Future Potential for AMD Intervention

As discussed earlier, the intervention in AMD involves multiple mechanisms, including antioxidant, anti-inflammatory, antimicrobial, and anti-aging effects. In this section, we highlight specific berries that have shown these properties in animal studies but still lack clinical validation, in particular in AMD, to further study the effects of that particular berry on AMD. We believe these berries hold potential for future research and clinical development, but more clinical trials are needed to confirm their efficacy in the prevention and management of AMD.

#### 4.2.1. Blueberry, *Vaccinium corymbosum*

Blueberries are rich in anthocyanins and other antioxidant polyphenols, such as phenolic acids and flavonoids. Preclinical studies have demonstrated that blueberry polyphenols can protect RPE cells from damage caused by light exposure and lipid peroxidation by reducing oxidative stress and inflammation [40,100]. Liu et al. reported that blueberry polyphenols ameliorate visible light and lipid-induced injury in RPE cells, with the phenolic acid-rich fraction showing superior efficacy in inhibiting cell death at a concentration of 10.0 μg/mL [100]. Furthermore, anthocyanin- and flavonoid-rich fractions were effective in preventing the expression of senescence-associated β-galactosidase and the overexpression of vascular endothelial growth factor (VEGF). The flavonoid-rich fraction also exhibited high activity in reducing phagocytosis and cellular oxidative stress.

In a separate study, Liu et al. found that anthocyanin components from wild Chinese blueberries could ameliorate light-induced retinal damage in pigmented rabbits [114]. Similarly, Huang et al. demonstrated the protective effects of blueberry anthocyanins against H₂O₂-induced oxidative injuries in human RPE cells [40]. Additionally, blueberry polyphenols were shown to inhibit visible light-induced lipid peroxidation of unsaturated fatty acids and exhibit anti-inflammatory effects in a light-induced rabbit retina injury model [101]. Anthocyanins in blueberries have also demonstrated neuroprotective potential by interacting with rhodopsin and enhancing antioxidative capacity in a rat model [102]. In a long-term study with a mean follow-up of 11 y, greater blueberry intake was reported to significantly reduce total AMD in women [99]. This is regarded as the first epidemiologic study that examined blueberry intake in the primary prevention of eye disease.

Moreover, specific components like pterostilbene have shown significant anti-inflammatory effects, potentially offering protection against retinal degeneration. For instance, Li et al. reported that pterostilbene from blueberries protects corneal epithelial cells from inflammation via an antioxidative pathway [103].

#### 4.2.2. Cranberry, *Vaccinium macrocarpon*

Cranberry, rich in proanthocyanidins (PACs), is known for its antimicrobial, antioxidant, and anti-inflammatory properties [104,115]. Preclinical studies have shown that a condensed tannin-containing fraction of cranberry juice exhibited better free radical scavenging activity and effectively protected ARPE-19 cells in a cell line model [105]. However, clinical trials specifically investigating cranberry’s impact on AMD are lacking, and further research is needed to confirm its potential benefits for eye health.

#### 4.2.3. Grapes and Grape Polyphenols (GPPs)

Grapes are highly valued for their rich content of bioactive compounds, and grape juice has been widely utilized around the world for its potent medicinal properties, including the promotion of ocular health. The health benefits of grape products are largely attributed to their diverse range of phytochemicals. Notably, grape polyphenols have shown significant potential in mitigating various eye conditions, such as macular degeneration, uveitis, cataracts, red eye, and diabetic retinopathy [106]. Natarajan et al. emphasized the role of grape polyphenols in supporting ocular health, highlighting their antioxidant, antimicrobial, anti-aging, anti-hypertensive, and anti-inflammatory properties.

For instance, proanthocyanidins derived from grape seeds exhibit strong antioxidant activity, which can help prevent macular degeneration and significantly delay the development of cataracts in ICR/f rats [106]. Additionally, Zhao et al. reported that grape skin extracts can inhibit apoptosis in a dose-dependent manner, suggesting that grape skin polyphenols could be promising candidates for the prevention and treatment of AMD [31].

Resveratrol, another important bioactive compound found in grapes, is discussed in detail in Section 3.4.

#### 4.2.4. Persimmon, *Diospyros kaki*

Persimmon leaves and fruits are rich in a variety of bioactive components, including proanthocyanidins, flavonoid oligomers, tannins, phenolic acids, and catechins [116]. Carotenoids and tannins are significant fractions, with β-carotene being the predominant carotenoid in persimmon fruit, followed by β-cryptoxanthin and α-carotene [116]. Due to its diverse phytochemical composition, persimmon and its products are considered effective in mitigating oxidative damage caused by ROS and in alleviating lifestyle-related conditions such as cardiovascular disorders and diabetes mellitus [116].

Persimmon may also have potential benefits for eye health. For instance, persimmon leaves have been reported to ameliorate MNU-induced retinal degeneration in mice [107]. Additionally, Ryul Ahn et al. found that ethanol extract from Diospyros kaki (EEDK) alleviated retinal degeneration resulting from nerve crush in a partial optic nerve crush mouse model. This protective effect was mediated by its antioxidant activities and modulation of apoptotic proteins, including PARP, p53, and caspase-3 [108].

#### 4.2.5. Gooseberry, *Emblica officinalis*

Gooseberry, commonly known as amla, is an important medicinal plant in both Ayurveda and Unani systems of medicine and is a key component in various herbal formulations, including patented drugs. The fruit of *Emblica officinalis* (EO) contains numerous bioactive compounds, such as ellagic acid, chebulinic acid, apigenin, gallic acid, quercetin, chebulagic acid, isostrictiniin, corilagin, methyl gallate, and luteolin [109]. Polyphenols, especially tannins and flavonoids, are primarily responsible for its major bioactivities. EO is a major ingredient in various health tonics and enhances medicinal efficacy through synergistic effects [110].

Every part of the *Emblica officinalis* plant is beneficial due to its extensive medicinal and pharmaceutical properties. The plant exhibits antioxidant, anti-inflammatory, anticancer, adaptogenic, antidiabetic, nootropic, antimicrobial, and immunomodulatory potential [109]. Additionally, EO helps prevent hyperlipidemia, osteoporosis, and other ailments [110].

EO may also play a cytoprotective role in AMD. Nashine et al. [111] reported that EO reduced the number of apoptotic and necrotic cells, attenuated amyloid-induced toxicity, prevented drusen deposit formation, and preserved mitochondrial and cellular health and function in human AMD retinal RPE cybrids. However, further clinical studies are necessary to validate these effects in human populations.

#### 4.2.6. Tart Cherry, *Prunus cerasus*

Tart cherry (*Prunus cerasus* L.) is rich in polyphenolic compounds, especially proanthocyanins, anthocyanins, and flavonols, and exhibits various antioxidant activities and health benefits [112]. Although research on the effects of tart cherry specifically for eye diseases is limited, some studies suggest potential benefits. These include the inhibition of mitochondrial apoptosis, modulation of aging and inflammatory conditions, and antioxidative and anti-inflammatory properties [112,113]. These findings suggest that tart cherry may hold promise for future exploration in the treatment of eye diseases.

## 5. Discussion

Although these berries have been proven or shown potential to be useful in the intervention of AMD, their clinical application often faces more complex challenges, including differences in bioavailability, varying biological activity, considerations for individual and combined use, and variability in individual responses.

### 5.1. Real Biological Activity and Bioavailability

The correlation between real biological activity and low bioavailability can often be complex. Though some berries show certain positive effects in vitro, they may not have shown the same effect in vivo. Factors such as poor solubility, instability in the gastrointestinal tract, rapid metabolism, and poor absorption can limit the amount of active substance that reaches the target tissues. Thus, a substance with strong biological activity but low bioavailability may require modifications, such as improved formulations or delivery methods, to enhance its effectiveness in real-world applications.

The bioavailability of different berry-derived phytochemicals significantly influences their efficacy in managing AMD. For example, anthocyanins from bilberries and black currants have low bioavailability due to rapid metabolism and excretion, as well as resveratrol and flavonoids [117,118]. In contrast, carotenoids such as lutein and zeaxanthin found in wolfberries have higher bioavailability and effectively accumulate in retinal tissue, reducing the risk of AMD [77]. Understanding these differences is crucial for developing effective recommendations and considering the incorporation of different phytochemicals and berries.

Understanding the impact of various food processing techniques on the stability and bioavailability of these compounds is essential. Techniques such as freeze-drying and infra-red drying can affect the antioxidant capacities of berries differently, offering insights into optimal processing methods that preserve and enhance the health benefits of these nutrients [119].

### 5.2. Clinical Outcomes and Targeted AMD Subtypes

Clinical evidence is more robust for some berries than others. For instance, bilberry’s role as an effective antioxidant has been extensively studied. Related clinical trials show improvements in visual fatigue and retinal function, while direct links to AMD prevention remain limited [72]. On the other hand, while blueberries and cranberries are known for their general health benefits, more specific clinical trials focusing on their role in AMD prevention are needed. Similarly, for persimmon, gooseberry, and tart cherry, further clinical studies are necessary to validate their effects in human populations. Considering all the available evidence, we believe that wolfberry can be the most promising candidate to direct further research to at first. Wolfberries have been confirmed to have a well-rounded function to benefit eye health (see Table 1) and demonstrated promise in increasing macular pigment optical density (MPOD), which correlates with a reduced risk of AMD [78].

Certain berries may be more effective against specific subtypes of AMD. For example, the high antioxidant and anti-inflammatory capacity of anthocyanin-rich bilberries might be particularly beneficial in preventing the progression of dry AMD by reducing oxidative damage and inflammation [120]. In contrast, wolfberry carotenoids, which protect against blue light damage, might be more suited for early intervention in individuals at risk of developing AMD [121].

### 5.3. Challenges in Clinical Translation

Despite promising preclinical data, translating these findings into clinical practice poses significant challenges. Though clinical evidence is more robust for some berries, these studies often fall short of establishing a direct link between berry consumption and AMD prevention or treatment. A key obstacle is the variability in bioavailability and individual metabolic responses, which complicates the standardization of dosages and treatment protocols. For better outcomes, individual and combined use of different berries, phytochemicals, and other nutrients is also being extensively tested [122]. Additionally, most clinical studies to date are limited in scale and scope, often focusing on short-term outcomes without addressing long-term efficacy and safety.

## 6. Conclusions

Berries, enriched with diverse antioxidative phytochemicals such as anthocyanins, flavonoids, carotenoids, and resveratrol, demonstrate promising potential in the prevention and management of age-related macular degeneration (AMD). Their antioxidative, anti-inflammatory, and anti-angiogenic properties not only support retinal health but also contribute to systemic health benefits. Preclinical studies and limited clinical trials indicate that compounds from berries like bilberry, aronia, wolfberry, and black currant can mitigate oxidative stress, reduce inflammation, and enhance visual function, thereby slowing AMD progression.

However, significant challenges remain in translating these findings into clinical practice. The low bioavailability of many berry-derived phytochemicals limits their effectiveness in vivo, necessitating further research into advanced delivery methods to enhance absorption and stability. Moreover, the current clinical evidence is largely preliminary, highlighting the need for well-designed, large-scale studies to establish standardized dosages, treatment protocols, and long-term efficacy. Understanding the synergistic effects of combining different berries or integrating them with other dietary components may also optimize therapeutic outcomes.

Future research should focus on mechanistic studies to elucidate the molecular pathways through which these bioactive compounds exert their protective effects. Additionally, exploring personalized nutrition strategies, which take into account individual genetic and environmental factors, may enhance the efficacy of berry-derived interventions in AMD management. By addressing these research gaps, the therapeutic potential of berries in ocular health can be fully realized, ultimately contributing to the development of dietary recommendations and nutraceutical formulations for AMD prevention.

## Figures and Tables

**Figure 1 antioxidants-13-01558-f001:**
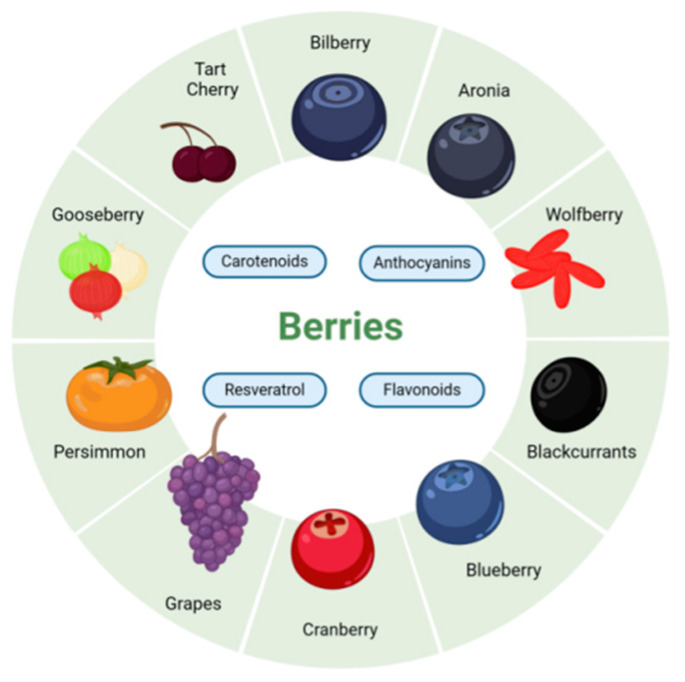
Berries and their phytochemical compounds with potential therapeutic effects. This figure presents a summary of berries and their associated phytochemicals, which have potential therapeutic effects in the context of age-related macular degeneration (AMD). Key compounds such as flavonoids, anthocyanins, carotenoids, and resveratrol are highlighted as primary bioactive agents within these berries, including blueberries, cranberries, blackcurrants, and gooseberries. These berries are central to the antioxidant properties of berries, making dietary interventions with these fruits increasingly relevant for the prevention and management of AMD and other ocular diseases.

**Figure 2 antioxidants-13-01558-f002:**
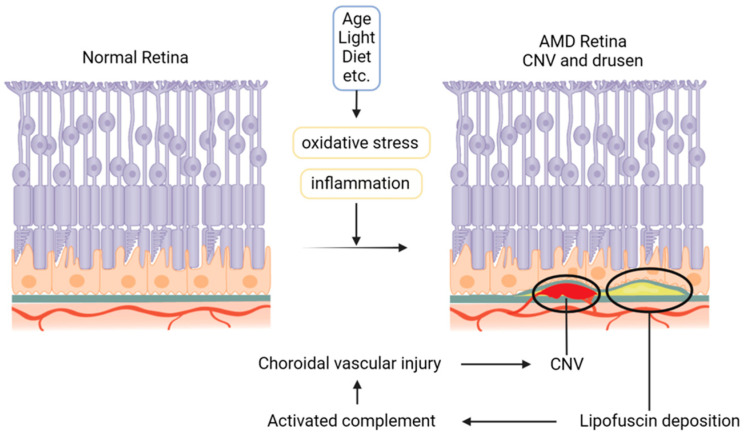
Pathophysiology of age-related macular fegeneration (AMD). The pathophysiological mechanisms behind the onset of age-related macular degeneration (AMD) are illustrated in this image. The diagram traces the progression from initial oxidative damage to the retina through the accumulation of drusen, advancing to geographic atrophy (GA), and culminating in choroidal neovascularization (CNV). Due to factors such as age, radiation, and diet, oxidative stress and chronic inflammation gradually accumulate, leading to lipofuscin deposition and activation of the complement system. These mechanisms are emphasized, offering insights into the key factors driving retinal degeneration in AMD.

**Figure 3 antioxidants-13-01558-f003:**
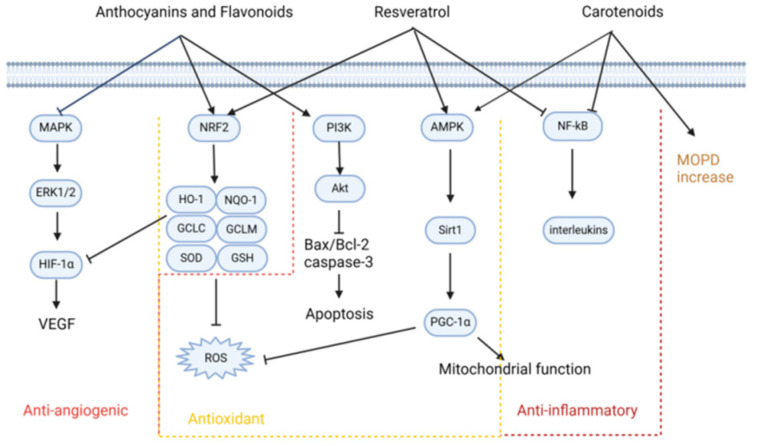
Protective mechanisms of phytochemicals in AMD. This figure illustrates the protective mechanisms employed by phytochemicals in mitigating the progression of age-related macular degeneration (AMD). It highlights the activation of the Nrf2 signaling pathway, which enhances antioxidant defense systems, and the regulation of apoptosis through the PI3K/AKT pathway. This figure also details how these compounds suppress oxidative stress in RPE cells and exert anti-inflammatory effects by inhibiting pathways such as TLR4/NF-κB and MAPK. Additionally, the anti-angiogenic properties of phytochemicals are showcased, specifically their role in downregulating the HIF-1α/VEGF/VEGFR2 signaling axis, which is crucial for preventing choroidal neovascularization.

**Table 2 antioxidants-13-01558-t002:** Experimental and clinical studies of berry phytochemicals in age-related macular degeneration.

Berries	Main Component	Animal Experiments			Clinical Trials Related to AMD		
		Main Functions	Dose	Reference	Result	Dose	Reference
Bilberry (*Vaccinium myrtillus* L.)	anthocyanins	anti-inflammation, antioxidation cleaning drusen, regenerating rhodopsin	500 mg/kg of bilberry extract, 4 days	[42]	upregulated oxidative stress defense enzymes in RPE cells	10^−6^–1.0 mg/mL of bilberry extract	[73]
			not available	[69]			
Aronia (*Aronia melanocarpa*)	anthocyanins	anti-inflammation, antioxidation, cleaning drusen	600 mg/kg of aronia extract, 28 days	[17]	not reported		
			600 mg/kg of aronia extract, 28 days	[74]			
Wolfberry (*Lycium barbarum* L.)	carotenoids, polysaccharides	anti-inflammation, antioxidation, anti-apoptosis, mitochondrial function, neuroprotective effects	listed in Table 1		increased macular pigment optical density, neuroprotective effects	28 g of wolfberry, five times weekly	[78]
						5 g (±7%) net weight of wolfberry granules	[79]
Black currant (*Ribes nigrum* L.)	anthocyanins	antioxidation, regenerating rhodopsin	not available	[69]	increased macular pigment optical density	100 mg, 167 mg, and 233 mg of blackcurrant extract	[98]
			25, 50, and 100 mg/kg of blackcurrant extract, 1 week	[95]			
Blueberry (*Vaccinium corymbosum*)	anthocyanins, polyphenols	anti-inflammation, antioxidation, anti-apoptosis, anti-phagocytosis, preventing the overexpression of VEGF, neuroprotective effects	5 μg/mL of blueberry anthocyanin extract (BAE), malvidin (Mv), malvidin-3-glucoside (Mv-3-glc), and malvidin-3-galactoside (Mv-3-gal) aqueous solution	[40]	reduced total AMD, but not visually significant AMD or cataracts in women	1 serving/week of blueberries (11.2 mg/d of anthocyanins)	[99]
			10 and 50 μg/mL of polyphenol mixture from blueberries	[100]			
			25 and 50 μg/mL of polyphenol mixture from blueberries	[101]			
			One milliliter of fortified blueberry extract, 7 weeks	[102]			
			5–20 μg/mL of pterostilbene (a natural component of blueberries)	[103]			
Cranberry (*Vaccinium macrocarpon*)	flavonoids	antimicrobial, antiadhesion, antioxidation, anti-inflammation	5–50 μg/mL of cranberry juice extract	[104,105]	not reported		
Grapes	polyphenols	antimicrobial, antioxidant, anti-inflammation	0.2–5 µg/mL of grape skin extracts	[31,106]	not reported		
Persimmon (*Diospyros kaki*)	flavonoids, polyphenols, carotenoids	anti-inflammation, antioxidation, anti-malignant, anti-apoptosis, neuroprotective effects	10, 50, 100 mg/kg of *Diospyros kaki* extract, 1 and 4 weeks	[107]	not reported		
			0.1, 1, and 10 µg/mL of *Diospyros kaki* extract	[108]			
Gooseberry (*Emblica officinalis*)	flavonoids, polyphenols	anti-inflammation, antioxidation, anti-apoptosis, mitochondrial function, cleaning drusen	10, 15, 20, and 25 mg/mL of *Emblica officinalis* extract	[109,110,111]	not reported		
Tart cherry (*Prunus cerasus* L.)	polyphenols	anti-inflammation, antioxidation, anti-apoptosis, mitochondrial function	480 mL of tart cherry juice, 12 weeks	[112]	not reported		
			200 μg/mL of tart cherry extract	[113]			

This table provides a comprehensive summary of various experimental and clinical studies investigating the potential therapeutic effects of berry-derived phytochemicals on age-related macular degeneration (AMD). The studies encompass different types of berries, including bilberry, aronia, and wolfberry, and examine their bioactive compounds, such as anthocyanins and other polyphenols. The table outlines the mechanisms through which these compounds exert their protective effects, the experimental models used, and the key findings that support the role of these phytochemicals in mitigating AMD progression. The evidence highlights the antioxidative, anti-inflammatory, and neuroprotective properties of these compounds, emphasizing their potential as natural dietary interventions for AMD management.

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
