# Peer review of "Berries and Their Active Compounds in Prevention of Age-Related Macular Degeneration"

_antioxidants, 2024, doi:10.3390/antiox13121558_

Round 1
Reviewer 1 Report
This is a thorough overview of Berries and their potential role in the prevention of AMD.
The authors really describe in depth the potential benefical aspects and how this is supported by the available research even though direct studies related to AMD are often lacking.
Here are a few suggestions to improve the manuscript:
1. The authors could elaborate more on the current insights of dietary habits en beneficial food intakes (such as the mediterranean diet and the use of vitamin supplements) in AMD since the available data in literature would support the potential role of berries in AMD since many berries contain elements that are also key players in these supplements and dietary patterns. I think this aspect is under-represented in the current version of the manuscript. (only briefly mentioned on page 14 lines 452-455).
2. The authors could consider to switch the order of the chapters 3 and 4 and start section 4 with a more elaborate story on the benefits of a healthy diet on AMD (as suggested above) and then continue with a summary of the major components from diet (and berries in particular) and their (potential) role in the prevention and management of AMD (that is already written down very nicely in this section!). Afterwards it would make more sense to look at the specific berries since these berries contain the compounds that you already have described to be beneficial in AMD. It would make the rationale on why to look specifically at berries much more clear.
3. In chapter 3 almost every section end with the same (almost rather obvious) statement that there is need for more research (in particular clinical trials in AMD) to further study the effects of that particular berry on AMD. It would make more sense to bundle this as a general remark at the end of the chapter instead of repeating the same statement in slightly different wordings after each individual berry.
4. It would be nice if the authors would speculate a bit more about which berry would be the most promising candidate to direct further research to at first (considering all the available evidence, including what is known on bioavailability). Now the paper ends with a general remark that further research is necessary without giving clear indication in which direction it should go.
Page 3, line 82: ref 22 is about food synergy and does not mention AMD, let alone the role of the choroid in AMD
Page 4, lines 141-143: vitamin supplementation has been know to significantly reduce risk of progression of AMD in the AREDS studies 1 and 2 and has been widely accepted as preventive 'treatment' for AMD, I would suggest to change the sentence accordingly.
Page 7, table 1. The layout of the table is messy, which makes interpretation difficult. In the text the authors state that "The mechanisms by which wolfberries may exert protective effects against AMD are 250 multifaceted and have been summarized in several studies (see Table 1)." However, table one summarizes results from several studies of which only one was focused on AMD. The results of this study are not mentioned in the text of the manuscript, even though this study focusses on an in vitro model of AMD. The relevance of the other studies in this table is unclear and also not elaborated on in the paper.
Page 17 , lines 637-639: it is incorrect to refer to the review paper of Mrowicka et al and say that they demonstrated the beneficial effect of adding lutein and zeaxanthin to the original AREDS1 formulation. This was demonstrated in the AREDS2 trial (ref 18 in this manuscript). The paper of Mrowicka et al is a review paper that does not bring new data.
Author Response
Dear Editors and Reviewers,
We appreciate your time and constructive feedback on our manuscript titled “Berries and Their Active Compounds in Prevention of Age-related Macular Degeneration”. We have carefully addressed all the comments and implemented the suggested revisions. Below is our detailed response to each point raised by the reviewers.
Reviewer 1
Comment 1: The authors could elaborate more on the current insights of dietary habits on beneficial food intakes (such as the Mediterranean diet and the use of vitamin supplements) in AMD since the available data in literature would support the potential role of berries in AMD since many berries contain elements that are also key players in these supplements and dietary patterns. I think this aspect is under-represented in the current version of the manuscript. (only briefly mentioned on page 14 lines 452-455).
Response 1: We have expanded the discussion of dietary habits, including the Mediterranean diet and vitamin supplements, and how these relate to the potential role of berries in AMD prevention (Line 144-149).
Comment 2: The authors could consider to switch the order of the chapters 3 and 4 and start section 4 with a more elaborate story on the benefits of a healthy diet on AMD (as suggested above) and then continue with a summary of the major components from diet (and berries in particular) and their (potential) role in the prevention and management of AMD (that is already written down very nicely in this section!). Afterwards it would make more sense to look at the specific berries since these berries contain the compounds that you already have described to be beneficial in AMD. It would make the rationale on why to look specifically at berries much more clear.
Response 2: According to your suggestion, the order of Chapters 3 and 4 has been switched. Also, they have been reorganized to begin with an introduction of the benefits of a healthy diet and then focus on specific berries (Line 141-152 and Line 463-473).
Comment 3: In chapter 3 almost every section end with the same (almost rather obvious) statement that there is need for more research (in particular clinical trials in AMD) to further study the effects of that particular berry on AMD. It would make more sense to bundle this as a general remark at the end of the chapter instead of repeating the same statement in slightly different wordings after each individual berry.
Response 3: Redundant remarks about the need for more research have been merged into a general comment in Line 652-655 and Line 792-795.
Comment 4: It would be nice if the authors would speculate a bit more about which berry would be the most promising candidate to direct further research to at first (considering all the available evidence, including what is known on bioavailability). Now the paper ends with a general remark that further research is necessary without giving clear indication in which direction it should go.
Response 4: We have included a discussion on which berries (e.g., bilberry, wolfberry) may be the most promising for future research based on current evidence (Line 778-800).
Comment 5: Page 3, line 82: ref 22 is about food synergy and does not mention AMD, let alone the role of the choroid in AMD
Response 5: We have removed this reference (Line 81).
Comment 6: Page 4, lines 141-143: vitamin supplementation has been known to significantly reduce risk of progression of AMD in the AREDS studies 1 and 2 and has been widely accepted as preventive 'treatment' for AMD, I would suggest to change the sentence accordingly.
Response 6: We have rewritten this sentence (Line 142-144).
Comment 7: Page 7, table 1. The layout of the table is messy, which makes interpretation difficult. In the text the authors state that "The mechanisms by which wolfberries may exert protective effects against AMD are 250 multifaceted and have been summarized in several studies (see Table 1)." However, table one summarizes results from several studies of which only one was focused on AMD. The results of this study are not mentioned in the text of the manuscript, even though this study focusses on an in vitro model of AMD. The relevance of the other studies in this table is unclear and also not elaborated on in the paper.
Response 7: (1) The layout of Table 1 has been edited. (2) This table aims to discuss the underlying mechanisms by which wolfberry prevents AMD. The table was revised and only studies related to AMD were retained. Some of these studies did not use AMD models but are related to the pathological mechanisms of AMD. (3) The corresponding text has been revised.
The revisions have been provided on Line 574-577 and Line 593-594, Table 1.
Comment 8: Page 17, lines 637-639: it is incorrect to refer to the review paper of Mrowicka et al and say that they demonstrated the beneficial effect of adding lutein and zeaxanthin to the original AREDS1 formulation. This was demonstrated in the AREDS2 trial (ref 18 in this manuscript). The paper of Mrowicka et al is a review paper that does not bring new data.
Response 8: We have rewritten this sentence (Line 331).

Reviewer 2 Report
This manuscript revises the ocular protective effects of berries and the associated phytochemicals from in vitro and in vivo studies, with a focus on age-related macular degeneration or AMD. There are limited reviews with a focus on berries and eye diseases. A review entitled “Edible berries: bioactive components and their effect on human health” was published in 2014 but this review discusses the protective effects of the substituents found in various kinds of berry and focuses on an overall health benefit. Sun et al. recently published a review article on the role of flavonoids in AMD (Biomedicine & Pharmacotherapy; Volume 159, March 2023, 114259). So, there may be some overlap between the current submission and Sun et al. study. Nevertheless, an updated review on the benefits of berries in AMD would add values to the research field of eye diseases. The review requires professional English editing. The authors must improve the presentation of figures and tables as described in detailed comments.
Line 21- In reference to this remark “A deeper understanding of these characteristics could enable the rational combination of berry-derived components to optimize therapeutic outcomes in AMD management.” What are the advantages of combining berry-derived components? What berry components are the authors suggesting? Lutein? Lutein plus flavonoids? Is this rationale based on the outcome of AREDS? The authors should discuss about which components are to be combined as a preventative treatment of AMD.
Line 17- Revise the phrase, the use of synthesizes is not appropriate
Line 50-what type of chronic diseases?
Line 71-The pathology of AMD is not entirely triggered by oxidative stress so it would be more appropriate to state that oxidative stress is one the major causes. Revise this phrase.
Figure 2 – what is Rays? What is ARMD? What is activated complement? Use uniform abbreviation for AMD. Are lines 134-139 part of the Figure legend? If so, revise it by removing this figure depicts…
Why did the authors only tabulate the major findings for Wolfberry? What about other berries? Provide a rationale about the importance of wolfberry among other berries in terms of the protective effects in the eyes.
Table 2. Summarising the findings on Wolfberry is a good idea but this table needs a major improvement to effectively summarise the study findings. Reformat the table so that the columns are aligned. Are the authors illustrating the protective functions of wolfberry in different animal models. If so, state the protective effect of wolfberry found from the studies. Is it anti-inflammatory, improve mitochondrial function or anti-apoptotic? What is loss of Δψm? Which tissues or cells were assessed? In the section about apoptosis, what effects did wolfberry have on the targets?
Line 333-Revise the title. What potentials, on eye diseases?
Please see comments for the authors.
Table 2. Does “not found” mean no clinical study has been done? If so, state not conducted or not reported. What diseases are the animal studies focused on? Which tissues or cells were assessed?
Should also discuss the recently published article on berry intake and eye diseases. J Nutr. 2024 Apr;154(4):1404-1413. doi: 10.1016/j.tjnut.2024.02.028. Intake of Blueberries, Anthocyanins, and Risk of Eye Disease in Women
Author Response
Dear Editors and Reviewers,
We appreciate your time and constructive feedback on our manuscript titled “Berries and Their Active Compounds in Prevention of Age-related Macular Degeneration”. We have carefully addressed all the comments and implemented the suggested revisions. Below is our detailed response to each point raised by the reviewers.
Reviewer 2
Comment 1: Line 21- In reference to this remark “A deeper understanding of these characteristics could enable the rational combination of berry-derived components to optimize therapeutic outcomes in AMD management.” What are the advantages of combining berry-derived components? What berry components are the authors suggesting? Lutein? Lutein plus flavonoids? Is this rationale based on the outcome of AREDS? The authors should discuss about which components are to be combined as a preventative treatment of AMD.
Response 1: Thanks for your comments. The advantage of combining berry-derived ingredients is lacking in evidence. Therefore, the relevant statements were removed. Based on your suggestions, we have proposed the recommended berries and their components. The revision has been provided on Line 21-23.
Comment 2: Line 17- Revise the phrase, the use of synthesizes is not appropriate
Response 2: The revision has been provided on Line 17.
Comment 3: Line 50-what type of chronic diseases?
Response 3: We have added the types of chronic diseases (Line 51).
Comment 4: Figure 2 – what is Rays? What is ARMD? What is activated complement? Use uniform abbreviation for AMD. Are lines 134-139 part of the Figure legend? If so, revise it by removing this figure depict.
Response 4: Figure 2 has been revised to use the uniform abbreviation for AMD. "Rays" has been revised to "Light". "Activated complement" means to activate the complement system. The meaning has been supplemented in the figure legends. The figure legends have been combined and simplified. The revision has been provided on Line 131-139, Figure 1.
Comment 5: Why did the authors only tabulate the major findings for Wolfberry? What about other berries? Provide a rationale about the importance of wolfberry among other berries in terms of the protective effects in the eyes.
Response 5: The mechanisms by which wolfberries exert protective effects against eye diseases have been extensively studied. Thus, we made a unique table for it. A rationale for emphasizing wolfberry and expanding the discussion on its relevance to AMD has been provided in Line 574-577.
Comment 6: Table 2. Summarising the findings on Wolfberry is a good idea but this table needs a major improvement to effectively summarise the study findings. Reformat the table so that the columns are aligned. Are the authors illustrating the protective functions of wolfberry in different animal models. If so, state the protective effect of wolfberry found from the studies. Is it anti-inflammatory, improve mitochondrial function or anti-apoptotic? What is loss of Δψm? Which tissues or cells were assessed? In the section about apoptosis, what effects did wolfberry have on the targets?
Response 6: We speculate that the comment is directed towards Table 1, which mainly summarizes the research on Wolfberry. Table 1 has been reformatted for better readability and includes additional details on study findings. The specific effects of wolfberry in different studies have been supplemented. "loss of Δψm" has been removed. The revisions have been provided on Line 593-594, Table 1.
Comment 7: Line 333-Revise the title. What potentials, on eye diseases?
Response 7: We have rewritten this sentence (Line 650).
Comment 8: Table 2. Does “not found” mean no clinical study has been done? If so, state not conducted or not reported. What diseases are the animal studies focused on? Which tissues or cells were assessed?
Response 8: We have changed the phrases in Line 751-752, Table 2.
Comment 9: Should also discuss the recently published article on berry intake and eye diseases. J Nutr. 2024 Apr;154(4):1404-1413. doi: 10.1016/j.tjnut.2024.02.028 intake of Blueberries, Anthocyanins, and Risk of Eye Disease in Women.
Response 9: We have added a discussion of the recent study on blueberry intake and eye diseases (J Nutr. 2024 Apr;154(4):1404-1413) (Line 677-680).

Round 2
Reviewer 2 Report
The authors have addressed my review report comments.
The authors have addressed my review report comments.